# OmniSAT: Compact Action Token, Faster Auto Regression

## Abstract

Existing Vision-Language-Action (VLA) models can be broadly categorized into diffusion-based and auto-regressive (AR) approaches: diffusion models capture continuous action distributions but rely on computationally heavy iterative denoising. In contrast, AR models enable efficient optimization and flexible sequence construction, making them better suited for large-scale pretraining. To further improve AR efficiency, particularly when action chunks induce extended and high-dimensional sequences, prior work applies entropy-guided and token-frequency techniques to shorten the sequence length. However, such compression struggled with *poor reconstruction or inefficient compression*. Motivated by this, we introduce an **O**mni **S**wift **A**ction **T**okenizer, which learns a compact, transferable action representation. Specifically, we first normalize value ranges and temporal horizons to obtain a **consistent representation** with B-Spline encoding. Then, we apply multi-stage residual quantization to the position, rotation, and gripper subspaces, producing **compressed discrete tokens with coarse-to-fine granularity** for each part. After pre-training on the large-scale dataset Droid, the resulting discrete tokenization shortens the training sequence by **6.8×**, and lowers the target entropy. To further explore the potential of *OmniSAT*, we develop a cross-embodiment learning strategy that builds on the unified action-pattern space and jointly leverages robot and human demonstrations. It enables scalable auxiliary supervision from heterogeneous egocentric videos. Across diverse real-robot and simulation experiments, OmniSAT encompasses higher compression while preserving reconstruction quality, enabling faster AR training convergence and model performance. Our project page is available at *OmniSAT*.

## 1 Introduction

Recently, VLA models (Ji et al., 2025; Black et al., 2024; Octo Model Team et al., 2023; Kim et al., 2024; O'Neill et al., 2024) have emerged as a promising route to general-purpose embodied intelligence, grounding visual–linguistic inputs into executable actions. Despite rapid advances in VLAs and robotic learning (Chi et al., 2023b; Liu et al., 2025; Intelligence et al., 2025), real-world deployment remains challenging due to the complexity of high-dimensional, long-horizon action spaces. Against this backdrop, current VLA methods largely fall into two categories: (i) **Diffusion-based** approaches (Black et al., 2024; Chi et al., 2023a) fit continuous action distributions via iterative denoising (Fig. 1 (a)). (ii) **Auto-regressive** (AR) approaches (Pertsch et al., 2025; Kim et al., 2024) model visual–language–action relations through next-token prediction (Fig. 1 (b)). Diffusion excels at continuous modeling but scales poorly due to costly denoising. By contrast, although AR training operates on discrete tokens and thus cannot directly parameterize the continuous action space, it enables efficient optimization and flexible sequence construction. These properties scale well with heterogeneous data sources, yielding a reliable and robust VLA backbone.

Recent works (Zhao et al., 2023; Kim et al., 2025) have proved that training on action chunks (*i.e.,* multi-step action segments) equips models with stronger temporal context understanding and reasoning. However, the long horizons expand the token sequences that slow AR optimization. To address this issue, common remedies either compress the trajectory via representation changes (Zhou et al., 2025) or shorten token streams with byte-pair encoding (Pertsch et al., 2025). The former is entropy-driven discretization that indeed shortens sequences but *incurs severe reconstruction error* at high compression ratios. The latter is built on token co-occurrence frequency statistics, where

Figure 1: **Comparison between Existing Approaches and OmniSAT.** (a) Diffusion-based policies require iterative denoising, limiting training efficiency and scalability. (b) AR policies train efficiently and support flexible sequence construction, but sacrifice fine-grained accuracy in continuous control. (c) OmniSAT amplifies AR efficiency through feasible high-rate compression while providing a unified token space that enables integration of heterogeneous datasets.

gains are capped by the domain gap between the training and target trajectories, leading to *weak out-of-domain generalization*. In summary, to enable efficient and effective AR training, a good action tokenizer should (i) *embed high-fidelity action trajectories*, preserving fine-grained execution details; and (ii) *provide sufficient compression*, enabling generative models to efficiently capture the correspondence between visual–linguistic contexts and executed actions over long horizons.

To this end, we introduce **OmniSAT**, an **O**mni **S**wift **A**ction **T**okenizer for high-quality compression. Specifically, OmniSAT first performs **consistency encoding** to normalize value ranges and convert variable-length trajectories into fixed-length control-point representations. It then applies multi-stage residual **quantization compression** separately to position, rotation, and gripper DoFs, producing discrete compressed codebook indices (*i.e.,* each index referencing a unified action pattern). As shown in Fig. 1 (c), our OmniSAT is first pretrained on the large-scale dataset Droid (Khazatsky et al., 2024) to distill the unified set of action patterns (*e.g.,* translation and grab). Subsequently, continuous actions are encoded as discrete token lists with an overall $\sim 6.8\times$ compression, while preserving millimeter-level reconstruction fidelity. To fully exploit AR scalability and OmniSAT transferability, we develop a cross-embodiment manipulation learning by mixing human egocentric videos (e.g., EgoDex (Hoque et al., 2025)) with robot demonstrations, strengthening the generalizability of the learned codebook action patterns. Across real-robot scenarios and diverse simulation benchmarks, OmniSAT achieves higher compression ratios and lower reconstruction errors than existing compression methods. When integrated into AR training, the reduced sequence length translates into faster convergence and stronger performance. To sum up, our contributions are threefold:

- We present **OmniSAT**, a unified two-stage tokenizer that yields a generalized action token space for scalable AR pretraining.
- We further explore **cross-embodiment manipulation learning** by incorporating human demonstrations, thereby more fully exploiting OmniSAT's potential for AR training.
- We demonstrate consistent gains in compression efficiency and downstream VLA performance across diverse real-robot and simulation benchmarks.

## 2 RELATED WORK

### 2.1 VISION-LANGUAGE-ACTION MODELS

The rapid progress of VLMs (Liu et al., 2024b; Dai et al., 2023; Bai et al., 2025; Team et al., 2025; Wu et al., 2025) in generalization and instruction following has catalyzed their use in robotics. Building on pretrained VLM backbones, large VLA models learn manipulation as next-action prediction from large-scale demonstrations (Ji et al., 2025; Kim et al., 2024; O'Neill et al., 2024). Despite improved instruction following and execution accuracy, VLAs still struggle with real-world tasks that demand long-horizon reasoning and out-of-distribution generalization. Current VLA methods

emphasize different trade-offs between execution fidelity and training efficiency. Diffusion-based approaches (Black et al., 2024; Intelligence et al., 2025) generate continuous trajectories via iterative denoising, delivering high precision but incurring substantial computational cost that limits scalability. In contrast, auto-regressive approaches (Pertsch et al., 2025; Kim et al., 2024) discretize actions into tokens and train with next-token prediction. While discretization may sacrifice some fine-grained continuity, AR training is markedly more efficient and supports flexible sequence construction, making it amenable to heterogeneous, large-scale datasets. Crucially, the effectiveness of AR-based VLAs hinges on mapping those continuous actions to discrete tokens, motivating the need for tokenizers that are both efficient and broadly compatible with diverse datasets and embodiments.

## 2.2 Discretized Action Representations

To enable AR training, prior work converts continuous control into discrete sequences so that policies can be optimized with token-level objectives (Szot et al., 2024a; Wen et al., 2024; Szot et al., 2024b). A common baseline is dimension-wise binning (Kim et al., 2024; Collaboration, 2023; Brohan et al., 2022), which is simple but sensitive to high-frequency variability and introduces quantization error. Structured alternatives address these limitations from complementary angles. Behavior Transformers (Shafiullah et al., 2022) cluster actions with $k$-means and predict residual offsets per head. VQ-BeT (Lee et al., 2024) encodes short chunks into a learned codebook via a residual VQ-VAE, improving expressiveness over binning, yet it assumes uniform input dimensionality and thus limits cross-embodiment transfer. Signal-compression methods shorten sequences to accelerate learning: FAST (Pertsch et al., 2025) applies a discrete cosine transform followed by byte-pair encoding (Sennrich et al., 2016), but the resulting variable-length tokens complicate batching, decoding, and AR training. BEAST (Zhou et al., 2025) represents trajectories with B-spline control points, but its reconstruction quality becomes poor at high compression ratios. In contrast, OmniSAT first aligns horizons by encoding entire action chunks into a fixed-length, high-fidelity representation, then performs DoF-wise residual quantization. They collectively yield high-quality reconstructions and a compact and transferable token space, enabling scalable AR training.

## 3 Methodology

### 3.1 Preliminaries

A VLA model aims to learn a policy $\pi_\theta$ that generates actions conditioned on visual observations $\boldsymbol{o}$ and language instructions $\boldsymbol{l}$. Formally, given a dataset $\mathcal{D} = \{(\boldsymbol{a}, \boldsymbol{l}, \boldsymbol{o})\}$, the AR training objective is to minimize the negative log-likelihood of actions:

$$\min_{\pi_\theta} \ \mathbb{E}_{(\boldsymbol{a},\boldsymbol{l},\boldsymbol{o})\sim\mathcal{D}} \big[ -\log \pi_\theta(\boldsymbol{a} \,|\, \boldsymbol{l}, \boldsymbol{o}) \big], \tag{1}$$

thus establishing a unified mapping between visual–linguistic context and action. To strengthen perception, instruction, and control coupling, our training target is not a single step but an action chunk $\boldsymbol{a} \in \mathbb{R}^{T \times d}$ (Zhao et al., 2023), where $T$ denotes the action horizon and $d$ denotes the DoF.

$$-\log \pi_\theta(\boldsymbol{a} \mid \boldsymbol{o}, \boldsymbol{l}) = \sum_{t=1}^{T} -\log \pi_\theta(\boldsymbol{a}_t \mid \boldsymbol{a}_{<t}, \boldsymbol{o}_t, \boldsymbol{l}), \tag{2}$$

where $\boldsymbol{a}_{<t} = (\boldsymbol{a}_1, \cdots, \boldsymbol{a}_{t-1})$ denotes for the action history. Since AR backbones operate on discrete sequences, we tokenize actions by discretizing continuous trajectories in each DoF $\boldsymbol{s} = \boldsymbol{a}_{1:T,i}$ with an action tokenizer $\mathcal{T}_a : \bar{\boldsymbol{q}} = \mathcal{T}_a(\boldsymbol{s})$, where $\boldsymbol{a} \in \mathbb{R}^{T \times d}$ is continuous and $\bar{\boldsymbol{q}} \in \{1, \cdots, K\}^L$ are integer indices with length $L$, where $K$ denotes the vocabulary size of action token.[1]

### 3.2 OmniSAT: Omni Swift Action Tokenizer

To achieve efficient and high-fidelity compression, we propose OmniSAT, which embeds continuous action trajectories as concise compressed quantization tokens. Specifically, given a trajectory with an arbitrary horizon, we first perform numerical and temporal normalization to obtain a fixed-length

---

[1] We use the notation $\bar{\cdot}$ for discretized quantities, e.g., $\bar{\boldsymbol{q}}$ for discrete action tokens.

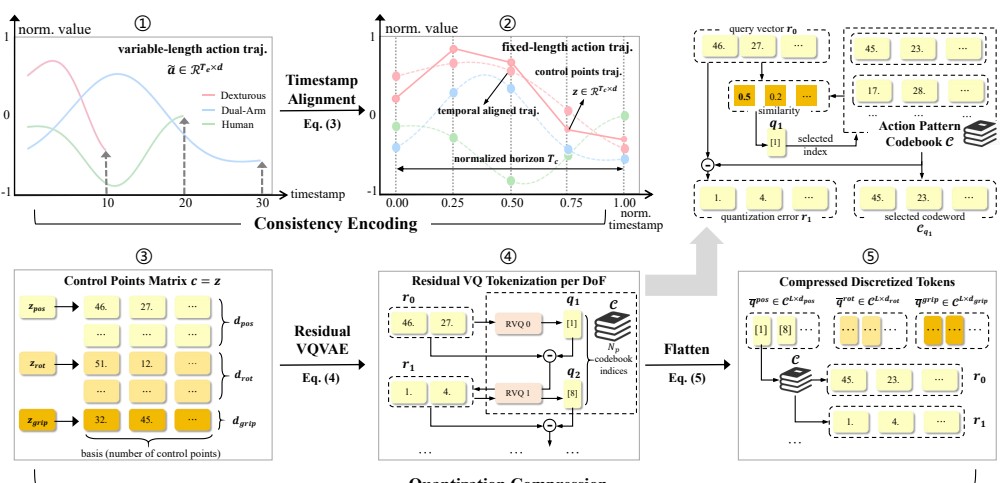

Figure 2: **Overview of OmniSAT Tokenization Pipeline.** **Consistency Encoding** converts variable-length trajectories into temporally aligned, fixed-length control-point representations via B-spline fitting. **Quantization Compression** splits control-point features into part groups (position, rotation, gripper) and applies residual vector quantization to obtain layerwise codebook indices. The selected indices are then flattened into final compact action-pattern tokens.

encoding. Then we quantize this normalized representation into discrete compressed tokens, shortening the target sequence and enabling efficient AR training. By coupling the two steps, OmniSAT enables **compact** (short sequences), **high-fidelity** (accurate reconstruction) compression tokenization. We elaborate on these steps below.

**Consistency Encoding.** To obtain a consistent action encoding across embodiments, we first apply robust per-DoF normalization for each dataset so that the 1st and 99th percentiles of each action dimension are mapped to $-1$ and $1$. In this way, we obatin the numerical normalized trajectory $\tilde{\boldsymbol{a}}_e \in \mathbb{R}^{T_e \times d}$ from embodiment $e$ with $T_e$ steps and $d$ DoFs (curves in Fig. 2 ①). Then, we encode them to a *fixed-length*, temporally aligned consistency representations $\boldsymbol{z} \in \mathbb{R}^{T_c \times d}$, where $T_c$ is the alignment length. Specifically, as illustrated in step ② (e.g., mapping the red dashed curve to the red solid curve), we encode the variable-length action $\tilde{\boldsymbol{a}}_e$ with the B-spline control points representation (each control point acts as a local "handle" shaping the nearby segment). Let $u_\tau = \frac{\tau-1}{T_e-1}$, $\tau = 1, \cdots, T_e$, be the normalized time grid for the samples. After this, the trajectories are normalized into the same horizon representation as dashed curves in step ②. Then we define $\Phi \in \mathbb{R}^{T_e \times T_c}$ as the uniform B-spline basis matrix evaluated on $\{u_\tau\}$. The resulting fixed-length control points representations $\boldsymbol{c} \in \mathbb{R}^{T_c \times d}$ are solved via ridge regression (detailed algorithm principles and solving process are provided in Sec. A.4 Prautzsch et al. (2002)):

$$\boldsymbol{c} = \arg\min_{\boldsymbol{c}} \|\Phi\boldsymbol{c} - \tilde{\boldsymbol{a}}_e\|_F^2 + \lambda\|\boldsymbol{c}\|_F^2, \tag{3}$$

where $\lambda > 0$ stabilizes the normal equations. The final control-point matrix $\boldsymbol{c}$ is taken as the consistency encoding representation output $\boldsymbol{z}$, serving as the input to the subsequent quantization.

**Quantization Compression.** Based on the fixed-length representations, extracting common action patterns across heterogeneous embodiments becomes feasible. To discretize these patterns precisely, we adopt a Residual Vector-Quantized VAE (Lee et al., 2024) technique. It approximates the feature vector $z$ through multi-layer residual quantization in a **coarse-to-fine manner**, which (i) **reduces distortion** for a general codebook and (ii) **yields higher effective compression** with fewer tokens at comparable fidelity. For clarity, we describe the single-DoF case $\boldsymbol{s} \in \mathbb{R}^{T_c}$: At each quantization layer $l \in \{1, \ldots, L\}$, the residual $r_{l-1}$ is quantized by selecting the closest codeword from the layer-specific codebook $\mathcal{C}^l = \{\mathcal{C}_i^l \in \mathbb{R}^T\}_{i=1}^K$, where $K$ is the codebook size. As shown in step ④ of Fig. 2, we initialize the first layer objective $\boldsymbol{r}_0 = \boldsymbol{s}$ and apply layer-wise recurrence:

$$q_l = \arg\min_{i \in [1,K]} \|\boldsymbol{r}_{l-1} - \mathcal{C}_i^l\|_F^2, \quad \boldsymbol{r}_l = \boldsymbol{r}_{l-1} - \mathcal{C}_{q_l}^l, \tag{4}$$

where $q_l$ is the selected codeword index at layer $l$. Here, the residual $\boldsymbol{r}_{l-1}$ represents the approximation error remaining from the previous layer, and subtracting $\mathcal{C}_{q_l}^l$ iteratively refines the reconstruc-

Figure 3: **OmniSAT for Cross-Embodiment Manipulation Learning.** The training pipeline has two phases: (i) **Tokenizer Pretraining**: OmniSAT is pretrained on heterogeneous human–robot datasets to learn a unified and compressed ($\times 6.8$) action token space; (ii) **Cross-Embodiment Fine-Tuning**: we construct mixed visual-action auto-regressive sequences over OmniSAT token space, enabling efficient and scalable fine-tuning through shorter sequences and lower target entropy.

tion. After $L$ layers, the quantized representation $\hat{s}$ and discrete token list $\bar{q}$ are given by:

$$\hat{s} = \sum_{l=1}^{L} \mathcal{C}_{q_l}^l, \quad \bar{q} = [q_1, q_2, \cdots, q_L]. \tag{5}$$

This yields the discrete tokenization for a single DoF. Repeating the above for every DoF, we obtain the full quantized representation. At inference time, we retrieve the codewords from each layer's codebook using the discrete indices $\bar{q}$ and sum them layer-wise to obtain $\hat{s}$ for each DoF. Finally, we stack the DoF-wise reconstructions to obtain the full reconstruction representation $\hat{z}$.

To further exploit the physical structure of actions and enhance codebook utilization, we partition the encoded representation $z$ into three semantically meaningful groups (Fig. 2 ③): (i) **Position** $z^{pos} \in \mathbb{R}^{d_{pos}}$, collecting all translational axes (*e.g.,* end-effector $x, y, z$). (ii) **Rotation** $z^{rot} \in \mathbb{R}^{d_{rot}}$, collecting all orientation parameters (e.g., roll, pitch, yaw). (iii) **Gripper** $z^{grip} \in \mathbb{R}^{d_{grip}}$, collecting the open–close states for gripper. Each group is quantized independently with its own residual quantization process and group-specific codebook $\mathcal{C}^{(g)}$, where $g \in \{\text{pos}, \text{rot}, \text{grip}\}$. This *part-level grouping* captures distinct motion semantics and produces discrete tokens $\bar{q}^{pos}$, $\bar{q}^{rot}$, and $\bar{q}^{grip}$, which are concatenated to form the final action token sequence: $\bar{q} = [\bar{q}^{pos}, \bar{q}^{rot}, \bar{q}^{grip}]$. In this way, part-group residual quantization achieves two objectives simultaneously: (i) *embedding high-fidelity action trajectories* by decomposing the approximation task into multiple smaller steps; and (ii) *yielding sufficient compressed discrete representation*, where the sequence of codebook indices directly serves as action tokens for downstream generative models.

**Training Objective.** To stabilize the codebooks and prevent collapse, we adopt three loss components. First, the **reconstruction loss** ensures that the quantized tokens preserve fidelity at both the representation and trajectory levels. Specifically, it consists of two terms: (i) reconstruction of the encoder features $z$ from the decoded quantized representation $\hat{z}$, and (ii) reconstruction of the original action trajectory $a$ from the control-point representation $z$ via B-spline decoding $\mathcal{B}(z)$:

$$\mathcal{L}_{\text{recon}} = \|z - \hat{z}\|_F^2 + \gamma \|a - \mathcal{B}(\hat{z})\|_F^2, \tag{6}$$

where $\mathcal{B}(\cdot)$ denotes the B-spline reconstruction operator (explained in the Sec. A.4) and $\gamma$ balances the two terms. Second, the **commitment loss** constrains the encoder outputs to remain close to specific codewords, while nudging the codebook vectors toward the encoder outputs:

$$\mathcal{L}_{\text{com}} = \|z - \text{sg}[\hat{z}]\|_F^2 + \|\text{sg}[z] - \hat{z}\|_F^2, \tag{7}$$

where $\text{sg}[\cdot]$ denotes the stop-gradient operator. The first term updates only the encoder (codewords are treated as constants); the second updates only the selected codewords (the encoder is treated as constant). Third, to strengthen the expressiveness of each quantization layer, we apply a **quantizer-layer dropout**. At training time, each residual layer $l$ is independently skipped with probability

$p=0.1$. Let $m_l$ be a Bernoulli binary mask with $1 - p$ distribution. The residual recursion and reconstruction processes become:

$$\boldsymbol{r}_l = \boldsymbol{r}_{l-1} - m_l\, \mathcal{C}_{q_l}^l, \quad \hat{\boldsymbol{z}}_{\mathbf{m}} = \sum_{l=1}^{L} m_l\, \mathcal{C}_{q_l}^l, \tag{8}$$

$$\mathcal{L}_{drop} = \|\boldsymbol{z} - \hat{\boldsymbol{z}}_{\mathbf{m}}\|_F^2, \tag{9}$$

where $\boldsymbol{m} = [m_1, \ldots, m_L]$. During training, we compute losses on the stochastically dropped reconstruction $\hat{z}_{\mathbf{m}}$; at inference, all layers are enabled ($m_l \equiv 1$). Building on the three objectives, we update with an exponential moving average for stable adaptation and obtain the overall objective:

$$\mathcal{L}_{\text{omnisat}} = \mathcal{L}_{\text{recon}} + \lambda_1\, \mathcal{L}_{\text{com}} + \lambda_2\, \mathcal{L}_{\text{drop}}, \tag{10}$$

where $\lambda_1, \lambda_2$ are weight coefficients (further analysis can be found at Tab. 9a and 9b).

### 3.3 CROSS-EMBODIMENT MANIPULATION LEARNING

Robotic datasets (Khazatsky et al., 2024; Bu et al., 2025; Hoque et al., 2025) exhibit substantial domain gaps, both in numerical distributions and action representations. Nonetheless, **OmniSAT** converts action trajectories from diverse datasets into discrete token representations within a shared action-pattern space. Building upon this foundation, we design a **cross-embodiment** training paradigm that consolidates demonstrations from both dual-arm robots and egocentric human demonstrations (Hoque et al., 2025) as shown in Fig. 3. Specifically, for each embodiment $e \in \mathcal{E}$, with instruction $\boldsymbol{l}^{(e)}$ and observations $\boldsymbol{o}^{(e)}$, we obtain per-frame visual tokens $\bar{\boldsymbol{v}}_t^{(e)}$ via the pretrained visual tokenizer $\tau_v$ in Wang et al. (2024) and per-frame action tokens $\bar{\boldsymbol{q}}_t^{(e)}$ via **OmniSAT** $\tau_a$:

$$\bar{\boldsymbol{v}}_t^{(e)} = \tau_v\big(\boldsymbol{o}_t^{(e)}\big), \quad \bar{\boldsymbol{q}}_t^{(e)} = \tau_a\big(\boldsymbol{a}_t^{(e)}\big). \tag{11}$$

We then form the frame-level packet prompt $\boldsymbol{u}$ and AR data stream $\boldsymbol{s}$ by concatenation:

$$\bar{\boldsymbol{u}}_t^{(e)} = \big[\bar{\boldsymbol{v}}_t^{(e)},\ \bar{\boldsymbol{q}}_t^{(e)}\big], \quad \bar{\boldsymbol{s}}^{(e)} = \big[\bar{\boldsymbol{u}}_1^{(e)}, \bar{\boldsymbol{u}}_2^{(e)}, \cdots\big]. \tag{12}$$

We treat multi-embodiment data as a weighted mixture and train a single AR objective across embodiments. For each embodiment $e \in \mathcal{E}$, we compute the standard next-token loss $\mathcal{L}_{\text{ar}}^{(e)}$ on its mixed visual–action token stream, then aggregate with mixture weights:

$$\mathcal{L}_{\text{ar}} = \sum_{e \in \mathcal{E}} \alpha^{(e)}\, \mathbb{E}_{\bar{\boldsymbol{s}}^{(e)} \sim \bar{\mathcal{S}}^{(e)}} \left[ -\frac{1}{|\bar{\boldsymbol{s}}^{(e)}|} \sum_{t=1}^{|\bar{\boldsymbol{s}}^{(e)}|} \log \pi_\theta\big(\bar{\boldsymbol{s}}_t^{(e)} \mid \bar{\boldsymbol{s}}_{<t}^{(e)},\ \boldsymbol{o}_t^{(e)},\ \boldsymbol{l}^{(e)}\big) \right],$$

where $\sum_e \alpha^{(e)} = 1$ and $\bar{\mathcal{S}}^{(e)}$ denotes the dataset of interleaved visual–action token streams for embodiment $e$. In this way, we tightly couple visual and action context at each step, enabling the policy to learn more robust cross-modal associations. (We detail the computation of the visual loss $\mathcal{L}_{vis}$ and action loss $\mathcal{L}_{act}$ in Eqs. 15 and 16.)

## 4 EXPERIMENTS

This section evaluates the effectiveness of *OmniSAT* in action tokenization and end-to-end VLA training. We study three research questions (RQ):

***RQ1: Advantages over existing action tokenizers.*** What advantages does OmniSAT provide relative to prior action tokenizers, *e.g.,* higher compression ratios, better reconstruction fidelity?

***RQ2: OmniSAT gains for AR training.*** How does OmniSAT benefit AR training, *e.g.,* dataset scalability and better model performance?

***RQ3: Design ablations of OmniSAT.*** How do the design choices in OmniSAT contribute to the imitation-learning performance, *e.g.,* loss settings and module configurations?

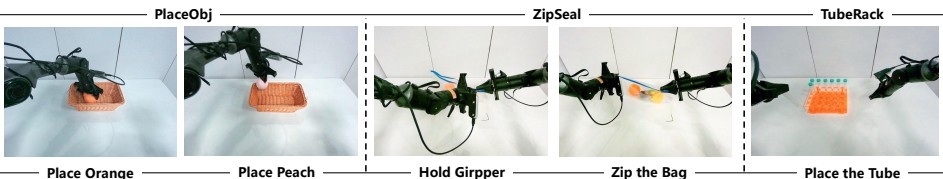

Figure 4: **Real-World Benchmark Visualizations.**

## 4.1 IMPLEMENTATION DETAILS

In real-world experiments, the backbone $\pi_\theta$ is a purely autoregressive Transformer with **8.5B** parameters, following the **Emu3** configuration (Wang et al., 2024) and world-model post-tuning (Wang et al., 2025). Images are tokenized with a spatial compression factor of $8\times$. In simulation, we use **Florence-2 Large** (Xiao et al., 2024) as the backbone with **0.77B** parameters. The original action horizon is set to 30 frames. We pretrain the tokenizer for 5 epochs to update the codebooks, using loss weights $\lambda_1{=}1.0$ and $\lambda_2{=}0.2$. More hyperparameter selection (*e.g.,* codebook size, compression chunk length, weight coefficients) can be found at Sec. A.2.

## 4.2 BENCHMARKS.

**Real-World Benchmarks.** As shown in Fig. 4, we evaluate three real-world tasks that emphasize different aspects of manipulation: PlaceObj, ZipSeal, and TubeRack (details can be found at Sec. A.1). **PlaceObj** is an instruction-grounded pick-and-place task, where the agent identifies the specified item among distractors, and places it into the correct place. **ZipSeal** is a contact-rich dexterity task, in which the agent aligns the edges of a resealable bag and closes it. **TubeRack** is a high-precision insertion task, where the agent grasps a test tube and inserts it into a rack slot.

**Simulation Benchmarks. LIBERO** (Liu et al., 2024a) comprises four task suites : spatial, object, goal, and long-horizon compositional. Each suite contains 10 robotic manipulation tasks, with 50 demonstrations provided for each task. **SimplerEnv** (Li et al., 2024b) reflects the performance of real-world policies by replicating physical dynamics and visual appearance, encompassing diverse variations in lighting, textures, and viewpoints. Detailed descriptions are provided in the Sec. A.3.

## 4.3 COMPARISON WITH EXISTING ACTION TOKENIZERS.

To address **RQ1**, we first evaluate **OmniSAT** against representative tokenizers along two axes (details in Sec. A.5): (i) **reconstruction quality** and (ii) **compression performance**. The evaluation is established on DROID (with 76k demonstration trajectories collected by Khazatsky et al. (2024)) dataset, using a 9:1 train-test split. For OmniSAT, we vary the quantization depth $L$ to trade off training efficiency against fidelity. Under this setup, we report test **MAE** and the **compression ratio** $R$ [2] paired with

Table 1: Comparisons of Compression Quality on DROID.

| Method | MAE ($\downarrow$) | R ($\uparrow$) |
|---|---|---|
| FAST | <**1e-5** | 3.7 |
| BEAST | 8.0e-2 | 4.6 |
| **OmniSAT-10** | 8.5e-4 | 4.9 |
| **OmniSAT-8** | 9.4e-4 | 6.8 |
| **OmniSAT-6** | 1.3e-3 | **8.1** |

BPE for all baselines. As summarized in Tab. 7, OmniSAT attains the strongest compression (from $\times4.9$ to $\times8.1$), exceeding FAST ($\times3.7$) and BEAST ($\times4.6$). We select $L{=}8$ as the default choice, which offers the best balance between reconstruction quality and compression.

To further illustrate the advantage brought by higher compression, we compare end-to-end learning dynamics on the LIBERO. We report the average performance results in Fig. 5 and the detailed sub-tasks (*i.e.,* Spatial, Object, Goal, and Long) results in Fig. 9. Results indicate that OmniSAT consistently attains higher success at equal training steps and reaches its plateau earlier at 2.5k steps (3.5k for FAST and 4k for BEAST), reflecting more **efficient optimization**. These results show that the tokenizer-level advantages (*e.g.,* higher compression and lower reconstruction error) can help in **faster convergence** in imitation learning, laying the groundwork for the following AR training.

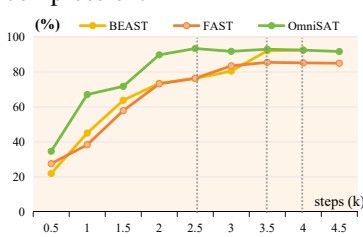

Figure 5: Training Convergence of Average Success on LIBERO.

---

[2]We explain the definition of evaluation metrics in Sec. A.5

Table 2: **Experimental Results for the LIBERO Benchmarks.** SR: Success Rate. Best results in each column are shown in bold.

| Model | Spatial | | Object | | Goal | | Long | | Average | |
|---|---|---|---|---|---|---|---|---|---|---|
| | SR (↑) | Rank (↓) | SR (↑) | Rank (↓) | SR (↑) | Rank (↓) | SR (↑) | Rank (↓) | SR (↑) | Rank (↓) |
| Octo (Octo Model Team et al., 2023) | 78.9% | 6 | 85.7% | 6 | 84.6% | 4 | 51.1% | 6 | 75.1% | 6 |
| OpenVLA (Kim et al., 2024) | 84.7% | 5 | 88.4% | 5 | 79.2% | 5 | 53.7% | 5 | 76.5% | 5 |
| SpatialVLA Qu et al. (2025) | 88.2% | 4 | 89.9% | 4 | 78.6% | 6 | 55.5% | 4 | 78.1% | 4 |
| FAST (Pertsch et al., 2025) | **96.4%** | 1 | 96.8% | 3 | 88.6% | 3 | 60.2% | 3 | 85.5% | 3 |
| BEAST (Zhou et al., 2025) | 92.9% | 3 | 97.5% | 2 | 93.1% | 2 | **86.4%** | 1 | 92.5% | 2 |
| **OmniSAT (ours)** | 94.1% | 2 | **98.7%** | **1** | **94.6%** | **1** | 86.0% | 2 | **93.4%** | **1** |

Table 3: **Evaluation on SimplerEnv–WidowX across diverse manipulation tasks.** Each task comprises two stages: picking up the specified object (**Grasp**), and putting the grasped object at the designated target location (**Success**).

| Model | Put Spoon | | Put Carrot | | Stack Block | | Put Eggplant | | Overall |
|---|---|---|---|---|---|---|---|---|---|
| | Grasp | Success | Grasp | Success | Grasp | Success | Grasp | Success | Success |
| RT-1-X Brohan et al. (2023) | 16.7% | 0.0% | 20.8% | 4.2% | 8.3% | 0.0% | 0.0% | 0.0% | 1.1% |
| Octo-Base Octo Model Team et al. (2023) | 34.7% | 12.5% | 52.8% | 8.3% | 31.9% | 0.0% | 66.7% | 43.1% | 16.0% |
| Octo-Small Octo Model Team et al. (2023) | 77.8% | 47.2% | 27.8% | 9.7% | 40.3% | 4.2% | 87.5% | 56.9% | 29.5% |
| RoboVLMs Li et al. (2024a) | 70.8% | 45.8% | 33.3% | 20.8% | 54.2% | 4.2% | 91.7% | 79.2 | 37.5% |
| BEAST (Zhou et al., 2025) | 66.7% | 41.7% | 37.5% | 25.0% | 50.0% | 20.8% | 87.5% | 75.0 | 37.5% |
| SpatialVLA Qu et al. (2025) | 20.8% | 16.7% | 29.2% | 25.0% | 62.5% | **29.2%** | 100% | **100%** | 42.7% |
| **OmniSAT(ours)** | **83.3%** | **58.3%** | **79.2%** | **37.5%** | **83.3%** | 29.2% | **100.0%** | 95.8% | **55.2%** |

## 4.4 MAIN RESULTS

To address **RQ2**, we assess our **OmniSAT** on three self-collected robot benchmarks (shown in Fig. 4) that span increasing task difficulty, and diverse simulation benchmarks.

### 4.4.1 REAL-WORLD EVALUATION

Beyond comparisons with baseline tokenizers (FAST and BEAST), we further introduce a hybrid training variant, **OmniSAT-M**ixed, designed to leverage the human demonstrations (from Egodex Hoque et al. (2025), which contains 300k episodes across 200 tasks) to further boost performance. During fine-tuning, **OmniSAT-M** interleaves human and robot data in the same token space, forming mixed batches at a **4:1** robot-to-human ratio. For each benchmark, we report success rates following fine-tuning on the corresponding dataset.

As shown in Figure 6, OmniSAT delivers clear gains, achieving 73%, 63%, 48% success rate, outperforming BEAST (63%, 45%, 23%) and Pi-FAST (38%, 18%, 38%). Adding human videos (OmniSAT-M) lifts the success rate at 80%, 66%, 58%, improving the average by **+6.7%** over OmniSAT (from 61.3% to 68.0%). Pi-FAST is notably weak on PlaceObj (38%) and ZipSeal (18%), indicating limited instruction-following and bimanual coordination capabilities. BEAST underperforms on TubeRack, indicating that its control-point approximation and reconstruction quality hinder learning precise manip-

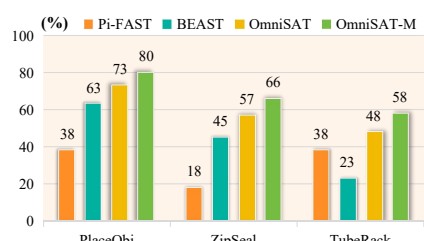

Figure 6: Real-World Evaluation.

ulation. Overall, OmniSAT tightens the relationship between visual–linguistic and action, and its unified action token space enables heterogeneous dataset scalability, strengthening fine-grained manipulation (ZipSeal and TubeRack) and multi-modal context understanding capabilities (PlaceObj).

## 4.5 SIMULATION BENCHMARKS EVALUATION

We train each policy on the full benchmark suite and evaluate on each sub-task, respectively. At test time, we run 50 and 24 rollouts per task with randomized initial states for LIBERO and SimplerEnv, respectively. On **LIBERO**, OmniSAT attains the best average success rate (93.4%), ranking #1 overall (Tab. 2). It sets the state of the art on Object (98.7%) and Goal (94.6%), while remaining competitive on Spatial (94.1%, rank 2) and Long (86.0%, rank 2). On **SimplerEnv**, OmniSAT

Table 4: **Ablations of OmniSAT Components.**

(a) Effects of $\mathcal{L}_{com}$ and $\mathcal{L}_{drop}$.

| $\mathcal{L}_{com}$ | $\mathcal{L}_{drop}$ | Spatial | Object | Goal | Long |
|---|---|---|---|---|---|
| ✓ | ✗ | 87.8 | 96.4 | **94.9** | 85.1 |
| ✗ | ✓ | 51.8 | 38.7 | 79.1 | 26.4 |
| ✓ | ✓ | **94.1** | **98.7** | 94.6 | **86.0** |

(b) Effects of CE and QC.

| CE | QC | Spatial | Object | Goal | Long |
|---|---|---|---|---|---|
| ✓ | ✗ | 92.9 | 97.5 | 93.1 | 86.4 |
| ✗ | ✓ | 93.8 | 98.3 | **94.8** | 83.2 |
| ✓ | ✓ | **94.1** | **98.7** | 94.6 | **86.0** |

Table 5: **Ablations of Backbone Selection and Vision Supervision.**

(a) Effect of Backbone Selection.

| Backbone | PlaceObj | ZipSeal | TubeRack |
|---|---|---|---|
| **Florence-Large** | 65% | 50% | 40% |
| **Emu3-Base** | **73%** | **57%** | **48%** |

(b) Effect of Vision Supervision in $\mathcal{L}_{ar}$.

| Vision Supervision | PlaceObj | ZipSeal | TubeRack |
|---|---|---|---|
| ✗ | 60% | **66%** | 33% |
| ✓ | **73%** | 57% | **48%** |

achieves the highest overall success (69.8%) and strong success rates across all tasks (Tab. 3). These results have proved the effectiveness and fidelity of the OmniSAT compression application.

## 4.6 ABLATION STUDY

To answer **RQ4**, we conduct ablations on the tokenizer design and downstream training choices, and provide detailed analyses in Sec. A.3 (including **weight coefficients, codebook sizes, and training speed, etc.**). These results collectively demonstrate the effectiveness of our design choices.

**Effects of Components in OmniSAT.** Tab. 4a shows that removing commitment loss causes enormous performance degradations, especially on Object (38.7%) and Long (26.4%), indicating the collapse phenomenon of codebook usage without the commitment loss. Without the dropout objective, we observe a decline in all subsets, excluding Goal. We conclude that the codebooks are overly reliant on certain codebook quantizers, resulting in poor generalization to the test set. Tab. 4b highlights complementary roles of the two stages: dropping Consistency Encoding (CE) mainly hurts Long (from 86.0 to 83.2%), indicating the normalization encoding representation helps in codeword quantization. While dropping the Action Quantization (AQ) mainly hurts Object and Goal (from 98.7%, 94.6% to 97.5%, 93.1%), which proves the effectiveness of RVQ-VAE compression. Using both together balances long-horizon stability with precise object and goal execution.

**Effects of Backbone and Objective.** In real-world tasks (Tab. 5a), Emu3-Base outperforms Florence-Large across all tasks, indicating that a stronger AR backbone (*i.e.,* larger context capacity and richer cross-attention) translates compression advantages into stable optimization, achieving higher success (from 51.7% to 59.3%). As shown in Tab. 5b, augmenting $\mathcal{L}_{AR}$ with visual-token prediction ($\mathcal{L}_{vis}$ in Eq. 15) boosts PlaceObj (from 60% to 73%) and TubeRack (from 33% to 48%), indicating that denser visual grounding improves instruction-conditioned reasoning and precise manipulation. The modest decline on ZipSeal (from 66% to 57%) likely reflects the deformable-object property: visual tokens emphasize appearance cues for multimodal understanding but offer weaker control signals for contact-centric actions. Overall, the compression benefits from OmniSAT materialize most when paired with a stronger AR backbone and modest visual supervision.

## 5 CONCLUSION

In this work, we introduced **OmniSAT**, an Omni Swift Action Tokenizer that enables scalable and efficient manipulation learning. OmniSAT factorizes trajectory modeling into two stages: (i) Normalizing horizon lengths and numerical distributions for obtaining consistency encoding representations. (ii) Discretizing position, rotation, and gripper actions into compressed, discretized tokens. The resulting compact token space unifies heterogeneous embodiments, providing a transferable foundation for auto-regressive VLA policy learning. Extensive experiments demonstrate that OmniSAT consistently and efficiently improves downstream policy performance and dataset scalability.

**Ethics Statement**   This work studies action tokenization and auto-regressive training for robot manipulation using public simulators (LIBERO, SimplerEnv, RoboCasa, RoboTwin2.0) and in-lab demonstrations on an AgileX bimanual platform (Fig. 7). Physical experiments were conducted in controlled spaces with interlocks, emergency stops, and conservative limits on speed and torque. No autonomous deployment in public settings was performed. We use consenting, non-identifying egocentric data and store it on secured servers following source licenses.

**Reproducibility Statement**   We will release PyTorch code for OmniSAT, Emu3 training scripts, configs per simulation benchmark, and pretrained tokenizer checkpoints, with a project page containing logs and instructions. Experiments use LIBERO Liu et al. (2024a), SimplerEnv-WidowX Li et al. (2024b), RoboCasa Nasiriany et al. (2024), RoboTwin2.0 Chen et al. (2025), DROID Khazatsky et al. (2024), and our 30 Hz real-robot datasets (400–600 frames; 90/10 split). Backbones: Emu3-Base (8.5B, decoder-only) for real-world training and Florence-2 Large (0.77B) for simulation. Tokenizer defaults: control-point length $T_c{=}8$, spline degree $4/0$ (pos/rot, gripper), codebooks $K^{\text{pos}}{=}256$, $K^{\text{rot}}{=}256$, $K^{\text{grip}}{=}64$, residual depth $L{=}8$ with quantizer-layer dropout $p_{\text{drop}}{=}0.1$, losses $\mathcal{L}_{\text{recon}}$ ($\gamma{=}1.0$), $\mathcal{L}_{\text{com}}$ ($\beta{=}0.2$), $\mathcal{L}_{\text{drop}}$ with $(\lambda_1, \lambda_2){=}(1.0, 0.2)$. Policy fine-tuning uses batch size 256, learning rate $5 \times 10^{-5}$, 10 epochs with 1-epoch warmup, bf16, sequence packing, and BPE vocab 2048. We follow official splits, fix random seeds, and report identical success metrics; runs were executed on $8{\times}$A800 GPUs.

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

## A APPENDIX

**Use of Large Language Models.** During manuscript preparation, we used a large language model (ChatGPT 5 Thinking) solely for language editing: polishing phrasing, improving logical flow between sentences, and checking typos. The technical ideas, algorithms, experiments, analyses, figures, and tables were conceived, implemented, and verified by the authors. We did not rely on the LLM for data generation, mathematical derivations, or empirical claims. All model-assisted edits were reviewed by the authors for factual accuracy and clarity before inclusion.

This supplementary material provides additional details on the proposed method and experimental results that could not be included in the main manuscript due to space constraints. Specifically, this appendix is organized as follows:

- Sec. A.1 describes the dataset and benchmark details.
- Sec. A.2 provides additional details on model architectures and training hyperparameters.
- Sec. A.3 presents supplementary experimental results and analysis.
- Sec. A.4 introduces the B-spline algorithm used in Action Normalization.
- Sec. A.5 illustrates the metrics used to evaluate performance

### A.1 BENCHMARKS

#### A.1.1 REAL-WORLD BENCHMARKS.

**DROID** (Khazatsky et al., 2024) is a large, in-the-wild manipulation corpus with 76k episodes (350 h) collected across 564 scenes in 52 buildings from 13 institutions, spanning 86 tasks/verbs. Each episode includes three synchronized RGB camera streams, depth, and calibration metadata (intrinsics/extrinsics), and natural-language instructions. Scenes cover kitchens, bathrooms, offices, bedrooms, labs, and more, with varied lighting, clutter, and viewpoints. We adopt the official splits and use DROID both to **pretrain** our tokenizer and to **evaluate compression/fidelity** (Tab. 7 under a fixed action horizon of 30 frames.

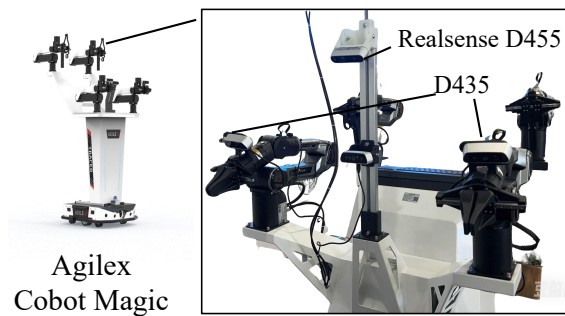

Figure 7: **Dual-arm data-collection platform.** We use an AgileX *Cobot Magic* base with two collaborative arms. RGB(-D) videos are captured from an overhead Intel RealSense D455 and wrist-mounted Intel RealSense D435i cameras; all streams are time-synchronized with joint-space commands at 30 Hz.

**Real-World Benchmarks.** We gather egocentric, goal-conditioned demonstrations on a dual-arm platform at 30 Hz (Fig. 7). Each trajectory logs joint-space commands for both arms and time-synchronized RGB(-D) video from (i) an overhead Intel RealSense D455 and (ii) wrist-mounted Intel RealSense D435i cameras. Typical clips contain 400–600 frames and paired language instructions. We summarize the benchmark objective and challenges in Table. 6 and present the visualization in Fig. 8:

#### A.1.2 SIMULATION BENCHMARKS.

**LIBERO** (Liu et al., 2024a) comprises four benchmark suites: LIBERO-Spatial, LIBERO-Object, LIBERO-Goal, and LIBERO-Long. Each suite consists of ten distinct tasks, designed to evaluate

Table 6: Task objectives centered with multirow; challenges split across three rows.

| Task | Objective | Challenge |
|------|-----------|-----------|
| **PlaceObj** | Place the named object into the basket. | Instruction grounding under distractors
Reliable table-top grasping
Precise placement into a confined receptacle |
| **ZipSeal** | Align the bag edges and close the zipper. | Contact-rich, deformable-object control
Coordinated bimanual hold–pull
Sustained force/pose control along the seal track |
| **TubeRack** | Pick up a test tube and insert it into a rack slot. | High spatial precision and tight tolerances
Stable transport without slip
Axial alignment and collision avoidance |

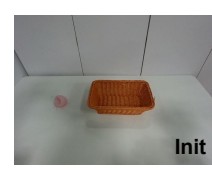 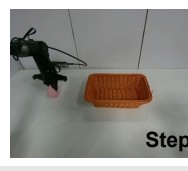 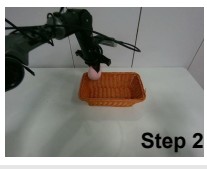 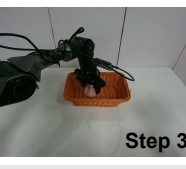 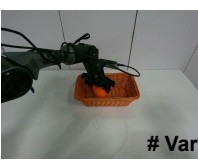

*PlaceObj:* "Place object on the table into the basket." (Step1) Grasp the object on the table surface. (Step2) Lift it up above the basket. (Step3) Release the object. (# Var) Task varies on the object category.

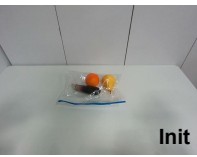 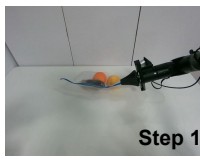 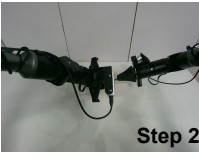 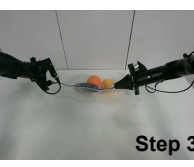 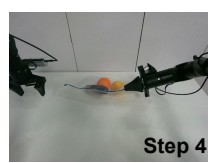

*ZipSeal:* "seal a plastic bag with zipper placed on the table." (Step1) Grasp one side of the bag. (Step2) Grasp the zipper tab with the other gripper. (Step3) Pulling the zipper along the sealing track. (Step4) Release the plastic bag.

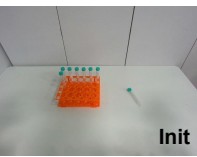 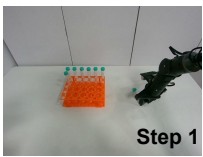 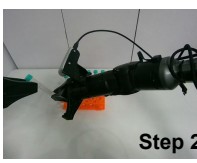 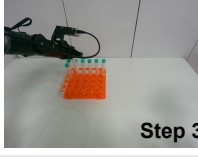 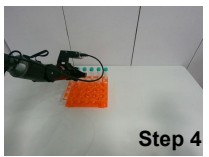

*TubeRack:* "Insert the tube on the table surface into an empty slot of the rack." (Step1) Pick up the tube. (Step2) Reorient the tube and pass it to the other gripper. (Step3) Insert the tube into the rack. (Step4) Release the tube.

Figure 8: **Real-world tasks and step-wise rollouts. PlaceObj**: grasp the instructed object, lift, and place it into the basket. **ZipSeal**: align both sides of a resealable bag, grasp the zipper tab, and close along the track. **TubeRack**: pick up a test tube, reorient, and insert it into the rack slot.

generalization along different axes—spatial layout, object identity, goal specification, and temporal composition. The tasks are implemented on a Franka Emika robot equipped with both a wrist-mounted camera and a third-person camera for visual observations. Each task is paired with natural-language instructions and 50 successful execution trajectories serving as demonstrations. We adopt the official evaluation protocol and report the average success rate across the tasks within each suite.

**SimplerEnv-WidowX** (Li et al., 2024b) is a simulated benchmark for evaluating how well robot manipulation policies transfer under controlled but significant domain shifts. Scenes vary in lighting, textures, color distributions, and camera pose; action and observation spaces match the WidowX platform. Its core design aims are to minimize visual and control gaps, ensuring realism, and making evaluation predictive of real-world behavior. We adopt the benchmark's "Put Spoon", "Put Carrot", "Stack Block", and "Put Eggplant" tasks and use the standard train/test splits and success criteria.

## A.2 MODEL ARCHITECTURE AND TRAINING HYPERPARAMETERS

**Backbones.** We use two decoder-only Transformer backbones in different settings. (1) **Florence-2 Large** Xiao et al. (2024) (simulation): a *0.77B*-parameter AR model that consumes interleaved visual–action packets. The visual front end is *DaViT*, i.e., Florence-2's dense vision transformer encoder, which produces per-frame visual embeddings fed directly to the AR stack alongside action tokens (no VQ tokenization). (2) **Emu3-Base** Wang et al. (2024) (real-world): an *8.5B*-parameter decoder-only Transformer used for hybrid training. Images are tokenized by Emu3's VQ tokenizer (spatial compression $8\times$) into discrete visual tokens that are interleaved with action tokens. Unless otherwise noted, the context length is *[ctx_len]*, the hidden size *[d_model]*, *[n_layers]* layers, *[n_heads]* attention heads, positional encoding *[type]*, and dropout *[p_drop]*. Mixed precision (bf16) and sequence packing are enabled.

**Input packing.** For each embodiment $e \in \mathcal{E}$ with system prompt $\boldsymbol{s}^{(e)}$, instruction $\boldsymbol{l}^{(e)}$, observations $\boldsymbol{o}_{1:T_e}^{(e)}$, and actions $\boldsymbol{a}_{1:T_e}^{(e)}$, we obtain per-frame tokens using the pretrained VQ visual tokenizer $\tau_v$ from Wang et al. (2024) and our action tokenizer $\tau_a$:

$$\bar{\boldsymbol{v}}_t^{(e)} = \tau_v\big(\boldsymbol{o}_t^{(e)}\big), \qquad \bar{\boldsymbol{q}}_t^{(e)} = \tau_a\big(\boldsymbol{a}_t^{(e)}\big). \tag{13}$$

We concatenate visual and action tokens at each step to form a frame-level packet and then build an AR stream under a causal mask:

$$\bar{\boldsymbol{u}}_t^{(e)} = \big[\bar{\boldsymbol{v}}_t^{(e)}; \bar{\boldsymbol{q}}_t^{(e)}\big], \qquad \bar{\boldsymbol{s}}^{(e)} = \big[\boldsymbol{s}^{(e)}, \boldsymbol{l}^{(e)}, \bar{\boldsymbol{u}}_1^{(e)}, \dots, \bar{\boldsymbol{u}}_{T_e}^{(e)}\big]. \tag{14}$$

**Causal masks and AR objective.** Let $\bar{\boldsymbol{s}}^{(e)} = [x_1^{(e)}, \dots, x_N^{(e)}]$ be the flattened token sequence (prompt, instruction, then packets). Define binary target masks for visual and action tokens:

$$m_j^{\text{vis},(e)} = \begin{cases} 1, & \text{if } x_j^{(e)} \text{ is a visual token} \\ 0, & \text{otherwise} \end{cases}, \qquad m_j^{\text{act},(e)} = \begin{cases} 1, & \text{if } x_j^{(e)} \text{ is an action token} \\ 0, & \text{otherwise.} \end{cases}$$

The model $\pi_\theta$ is a decoder-only AR Transformer Wang et al. (2024) trained with a standard next-token loss under a causal attention mask. We compute per-type cross-entropy losses by selecting targets with the masks:

$$\mathcal{L}_{\text{vis}}^{(e)} = -\frac{1}{\sum_j m_j^{\text{vis},(e)}} \sum_{j=1}^{N} m_j^{\text{vis},(e)} \; \log \pi_\theta\big(x_j^{(e)} \,\big|\, x_{<j}^{(e)}\big), \tag{15}$$

$$\mathcal{L}_{\text{act}}^{(e)} = -\frac{1}{\sum_j m_j^{\text{act},(e)}} \sum_{j=1}^{N} m_j^{\text{act},(e)} \; \log \pi_\theta\big(x_j^{(e)} \,\big|\, x_{<j}^{(e)}\big). \tag{16}$$

The hybrid visual–action objective averages across embodiments with weights $\lambda_{\text{vis}}, \lambda_{\text{act}} > 0$:

$$\mathcal{L}_{\text{AR}} = \frac{1}{|\mathcal{E}|} \sum_{e \in \mathcal{E}} \Big(\lambda_{\text{vis}} \mathcal{L}_{\text{vis}}^{(e)} + \lambda_{\text{act}} \mathcal{L}_{\text{act}}^{(e)}\Big). \tag{17}$$

This implements a single AR objective in which the causal mask enforces left-to-right conditioning, while the target masks decide whether a given position contributes to the visual or action loss.

**OmniSAT configuration.** OmniSAT comprises two stages: Action Normalization and Action Quantization. We provide the hyperparameters in each stage, respectively.

- **Action Normalization (B-spline encoding).** We apply per-channel robust normalization (1st/99th percentiles) and fit B-splines to each DoF. Degrees: 3 for position/rotation, 0 for gripper. The aligned control-point length (per DoF) is set to $T_a = 30$ (default), chosen for millimeter-level reconstruction. Ridge coefficient $\lambda = 1\text{e-}3$. The resulting fixed-length control-point matrix $z \in \mathbb{R}^{T_a \times d_e}$ is forwarded to the quantizer.

- **Action Quantization (part-group residual VQ-VAE).** We quantize $z$ in three part groups with independent codebooks: position $K^{\text{pos}}$ of size 256, rotation $K^{\text{rot}}$ of size 256, gripper $K^{\text{grip}}$ of size 64. Residual levels $L = 8$ per group; codeword dimensionality $d_c = 30$ (consistent to the aligned control-point length). Codebooks use EMA updates with decay $\alpha = 0.99$. Losses: reconstruction $\mathcal{L}_{\text{recon}}$ with coefficients $\alpha = 0.2$, commitment $\mathcal{L}_{\text{com}}$, and dropout regularization $\mathcal{L}_{\text{drop}}$; total
$$\mathcal{L}_{\text{vae}} = \mathcal{L}_{\text{recon}} + \lambda_1 \mathcal{L}_{\text{com}} + \lambda_2 \mathcal{L}_{\text{drop}}, \quad (\lambda_1, \lambda_2) = (1.0, 0.2).$$

**Token budgets and compression.**   Original action horizon is $30\,\text{Hz}$. Real-robot streams are compressed to $L{=}8$ action tokens per second ($3.75\times$). During AR training, we apply BPE to the interleaved stream (vocab 2048), yielding an additional $\sim 1.8\times$ reduction (overall $\sim 6.75\times$).

**Training hyperparameters.**   Tokenizer pretraining runs for 5 epochs with AdamW (betas $(0.9, 0.95)$, weight decay 0.1), batch size 8192, and learning rate 2e-4. Hybrid policy fine-tuning uses AdamW with learning rate $\eta = 5e{-}5$, batch size 256 on $8\times$A800, cosine decay with 1 warmup epoch and total 10 epochs. Visual loss weighting $\omega_{\text{vis}} = 0.3$ (when enabled).

## A.3   ADDITIONAL EXPERIMENTS

This section expands on OmniSAT with hyperparameter sweeps, efficiency measurements, and simulation results.

**Chunk length $L$.**   We ablate the aligned control–point length $T_a{=}L \in \{4, 6, 8, 10, 12\}$ on DROID. Shorter $L$ increases compression but can underfit fast motions; larger $L$ improves fidelity but lengthens token streams. As summarized in Tab. 7, $L{=}8$ offers the best trade-off: millimeter-level MAE with a strong end-to-end compression ratio, and it is therefore our default in all main results.

Table 7: Comparisons of Compression and Quality on DROID.

| Method | MAE ($\downarrow$) | R ($\uparrow$) |
|---|---|---|
| **FAST** | $<$**1e-5** | 3.7 |
| **BEAST** | 8.0e-2 | 4.6 |
| **OmniSAT-12** | 7.0e-4 | 3.8 |
| **OmniSAT-10** | 8.5e-4 | 4.9 |
| **OmniSAT-8** | 9.4e-4 | 6.8 |
| **OmniSAT-6** | 1.3e-3 | 8.1 |
| **OmniSAT-4** | 3.1e-2 | 11**.2** |

**Codebook size.**   We vary the per part-group vocabulary sizes $K^{\text{pos}}, K^{\text{rot}}, K^{\text{grip}}$ on DROID. Increasing codebook size reduces MAE up to a saturation point, after which gains are marginal while AR sequences grow longer due to weaker BPE reuse. The default $(256, 256, 64)$ balances fidelity and throughput.

Table 8: Effects of Part-Group Codebook Size.

| $K^{\text{pos}}$ | $K^{\text{rot}}$ | $K^{\text{grip}}$ | **MAE** ($\downarrow$) |
|---|---|---|---|
| 512 | 512 | 128 | **9.1e-4** |
| 512 | 512 | 64 | **9.1e-4** |
| 256 | 256 | 64 | 9.4e-4 |
| 256 | 256 | 32 | 9.6e-4 |
| 128 | 128 | 32 | 1.1e-3 |

**Training efficiency.**   With the same backbone and batch size (256), OmniSAT trains faster than both baselines: 12.56k ms/batch versus 13.73k ms/batch for FAST and 14.47k ms/batch for BEAST. This corresponds to 8% and 13% lower step time, respectively. The gain stems from emitting fewer action tokens per second with fixed-length packets, which reduces sequence length and improves compute utilization during AR training.

**Effects of weight coefficients $\lambda_1$ and $\lambda_2$**   Table 9a and Table 9b report the results of varying the two weight coefficients $\lambda_1$ and $\lambda_2$ in our loss function. For $\lambda_1$, we observe that increasing the weight from $0.0$ to $1.0$ gradually reduces the reconstruction error, achieving the lowest MAE at $\lambda_1 = 1.0$ ($9.4 \times 10^{-4}$). Further enlarging $\lambda_1$ beyond $1.0$ leads to a slight degradation, suggesting that excessively emphasizing this term harms the balance of optimization. A similar trend is found for $\lambda_2$, where the error decreases as $\lambda_2$ increases from $0.0$ to $0.2$, reaching the optimal MAE at $\lambda_2 = 0.2$ ($9.4 \times 10^{-4}$). However, larger values (e.g., $0.4$ and $0.5$) cause substantial performance drops, indicating over-regularization. These results highlight that both coefficients require careful tuning, and moderate values yield the best trade-off between stability and accuracy.

Table 9: **Effects of weight coefficients $\lambda_1$ and $\lambda_2$.**

(a) Effect of $\lambda_1$.

|  | 0.0 | 0.6 | 0.8 | 1.0 | 1.2 | 1.4 |
|---|---|---|---|---|---|---|
| **MAE** | 1.8e-3 | 1.2e-3 | 1.1e-3 | **9.4e-4** | 1.0e-3 | 1.3e-3 |

(b) Effect of $\lambda_2$.

|  | 0.0 | 0.1 | 0.2 | 0.3 | 0.4 | 0.5 |
|---|---|---|---|---|---|---|
| **MAE** | 1.7e-3 | 1.0e-3 | **9.4e-4** | 9.6e-4 | 7.2e-3 | 3.5e-2 |

**LIBERO per-suite results.** Across LIBERO's four suites (Spatial, Object, Goal, Long), OmniSAT delivers state-of-the-art average performance. It matches or surpasses the strongest prior in *Object* and *Goal*, remains competitive on *Long*, and shows consistently faster convergence—reaching its performance plateau with fewer training steps than BEAST and FAST. Per-suite learning curves are shown in Fig. 9, which also highlights OmniSAT's early gains and stable end performance.

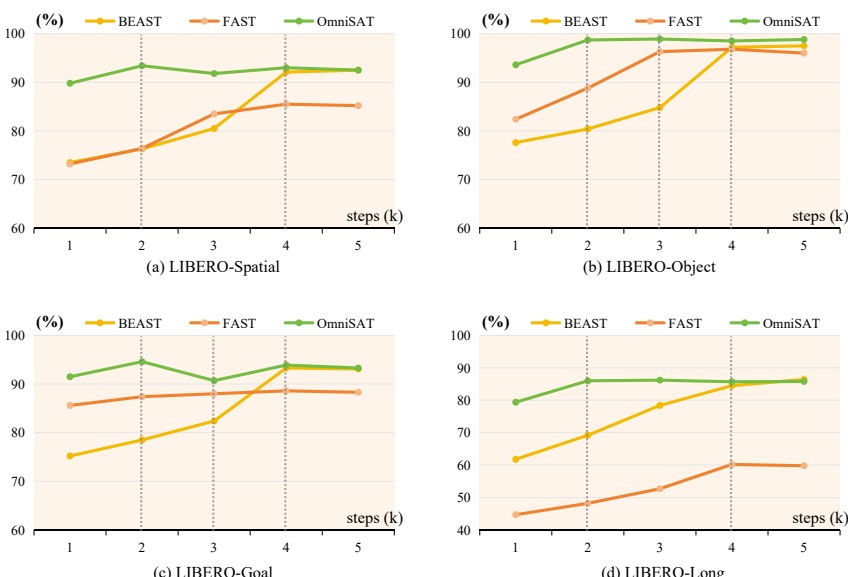

Figure 9: Training Convergence on LIBERO Benchmarks.

### A.4 B-SPLINE ALGORITHM THEORY

**B-spline basis.** A univariate B-spline of degree $p$ is defined over a nondecreasing knot vector $\mathcal{U} = \{u_0, \ldots, u_M\}$ with $M = N + p$ for $N$ control points. The $p$-degree basis functions $\{N_{i,p}(u)\}_{i=0}^{N-1}$ are given by the Cox–de Boor recursion:

$$N_{i,0}(u) = \begin{cases} 1, & u_i \leq u < u_{i+1}, \\ 0, & \text{otherwise,} \end{cases} \tag{18}$$

$$N_{i,p}(u) = \frac{u - u_i}{u_{i+p} - u_i} N_{i,p-1}(u) + \frac{u_{i+p+1} - u}{u_{i+p+1} - u_{i+1}} N_{i+1,p-1}(u), \tag{19}$$

with the convention $0/0 := 0$. We employ *uniform clamped* knots (first and last knots repeated $p+1$ times), which ensure interpolation at the endpoints and stable evaluation.

**Trajectory model.** A B-spline trajectory of degree $p$, parameterized over $u \in [0, 1]$ and defined by control points $\boldsymbol{c} = [c_0, \ldots, c_{N-1}]^\top$, is given by

$$y(u) = \sum_{i=0}^{N-1} c_i N_{i,p}(u), \qquad u \in [0, 1]. \tag{20}$$

Given a normalized action sequence $a_{1:T} = [a_1, \ldots, a_T]$ with length $T$, the objective is to construct a B-spline trajectory $y(u)$ that approximates the action sequence. A linear transformation maps the action timestep to the parametric space of the spline trajectory, where the sampled grid is defined as

$$u_\tau = \frac{\tau - 1}{T - 1}, \qquad \tau = 1, \ldots, T. \tag{21}$$

**Design matrix and least squares.** To make the B-spline trajectory on the sampled grid $y(u)_{1:T}$ closely matches the action sequence $a_{1:T}$, the control points are obtained by minimizing the least-squares error

$$\boldsymbol{c}^{\star} = \arg\min_{\boldsymbol{c}} \ \big\|y(u)_{1:T} - a_{1:T}\big\|_2^2. \tag{22}$$

We further formalize the precomputed B-spline basis functions of $y(u)$ into the B-spline design matrix $\boldsymbol{\Phi} \in \mathbb{R}^{T \times N}$, where

$$\Phi_{\tau,i} = N_{i,p}(u_\tau), \qquad \tau = 1,\dots,T, \ \ i = 0,\dots,N-1. \tag{23}$$

Under this formulation, estimating the B-spline trajectory parameters reduces to a standard least-squares optimization problem:

$$\boldsymbol{c}^{\star} = \arg\min_{\boldsymbol{c}} \ \big\|\boldsymbol{\Phi}\boldsymbol{c} - a_{1:T}\big\|_2^2, \quad \text{(unweighted)} \tag{24}$$

$$\boldsymbol{c}^{\star} = \arg\min_{\boldsymbol{c}} \ \big\|\boldsymbol{W}^{1/2}(\boldsymbol{\Phi}\boldsymbol{c} - a_{1:T})\big\|_2^2 \quad \text{(weighted)}, \tag{25}$$

where $\boldsymbol{W} \succeq 0$ can emphasize specific timestamps (e.g., contacts).

**Ridge regularization.** To improve numerical stability and avoid overfitting when the number of control points $N$ is large or the samples are noisy, we introduce an $\ell_2$ penalty, parameterized by a regularization coefficient $\lambda$ ($\lambda > 0$):

$$\boldsymbol{c}^{\star} = \arg\min_{\boldsymbol{c}} \ \big\|\boldsymbol{\Phi}\boldsymbol{c} - a_{1:T}\big\|_2^2 + \lambda\|\boldsymbol{c}\|_2^2 \ = \ (\boldsymbol{\Phi}^{\top}\boldsymbol{\Phi} + \lambda\boldsymbol{I})^{-1}\boldsymbol{\Phi}^{\top}a_{1:T}. \tag{26}$$

In practice we precompute $\boldsymbol{\Phi}$ from equation 23 and solve equation 26 via a Cholesky factorization of $(\boldsymbol{\Phi}^{\top}\boldsymbol{\Phi} + \lambda\boldsymbol{I})$. This batched procedure incurs only a minor computational overhead, typically on the order of a few milliseconds.

**From 1-DoF to multi-DoF.** For a $D$-DoF action sequence $\boldsymbol{a}_{1:T} \in \mathbb{R}^{T \times D}$, we fit each DoF independently using the same time grid $u_{1:T}$ and design matrix $\boldsymbol{\Phi}$:

$$\boldsymbol{c}_d^{\star} = (\boldsymbol{\Phi}^{\top}\boldsymbol{\Phi} + \lambda\boldsymbol{I})^{-1}\boldsymbol{\Phi}^{\top}a_{1:T}^{(d)}, \qquad d = 1,\dots,D. \tag{27}$$

Stacking $\{\boldsymbol{c}_d^{\star}\}_{d=1}^{D}$ yields the control-point matrix

$$\boldsymbol{C} = \begin{bmatrix} (\boldsymbol{c}_1^{\star})^{\top} \\ \vdots \\ (\boldsymbol{c}_D^{\star})^{\top} \end{bmatrix} \in \mathbb{R}^{D \times N}, \tag{28}$$

which serves as the fixed-length representation. Reconstruction at any $u$ follows from equation 20 using the corresponding row of $\boldsymbol{\Phi}$.

**Choice of degree, knots, and control points.** We use clamped uniform knot sequences, with spline degree set to $p=4$ for the smooth, high-fidelity representation of position and rotation trajectories, and $p=0$ for the near piecewise-constant representation of gripper signals. The number of control points $N$ balances reconstruction accuracy against representation compactness. In practice, we choose $N \ll T$ to align variable horizons into a common fixed length while retaining millimeter-level reconstruction.

## A.5 METRICS.

**Reconstruction MAE.** Given a test trajectory $\boldsymbol{a}_{1:T} \in \mathbb{R}^{T \times d}$ and its reconstruction $\hat{\boldsymbol{a}}_{1:T}$ obtained by decoding tokens (RVQ $\rightarrow$ control points $\rightarrow$ B-spline), we compute

$$\text{MAE} = \frac{1}{T \times d} \sum_{t=1}^{T} \sum_{k=1}^{d} \big|a_{t,k} - \hat{a}_{t,k}\big|,$$

after denormalizing to the original units per DoF. When reporting a single number, we average MAE over all test trajectories and DoFs.

**Compression ratio** ($R$). We measure end-to-end sequence compression relevant for AR training. Let $T$ be the original action horizon (continuous timesteps) and $L$ be the number of action tokens produced by OmniSAT (concatenating part group indices). After applying BPE, the effective token length becomes $L_{\text{BPE}}$. The compression ratio is

$$R \;=\; \frac{T}{\mathbb{E}[L_{\text{BPE}}]},$$

i.e., average timesteps per (BPE merged) token over the test set. We compute $R$ identically from their token streams with the same BPE vocabulary (2048).

