# OpenReview forum: "OmniSAT: Compact Action Token, Faster Auto Regression"
_ICLR.cc/2026/Conference — ICLR 2026 Conference Withdrawn Submission_

### Official Review · Reviewer_GBxv · 2025-10-15

**Soundness:** 3
**Presentation:** 2
**Contribution:** 2
**Rating:** 4
**Confidence:** 4

**Summary:**

The paper proposes OmniSAT, a two-stage tokenizer for robot action sequences intended to make autoregressive (AR) VLA training more efficient:
- Spline-based alignment (Consistency Encoding): fit B-spline control points to each action DoF to convert variable-length trajectories into a fixed-length representation using ridge-regularized least squares.
- Quantization (Action Quantization): residual VQ with separate codebooks for position, rotation, and gripper; flattened indices are then used as discrete action tokens for AR training

Empirically, the paper reports a high compression ratio at modest reconstruction error on a large pretraining corpus, faster AR convergence, and competitive downstream performance on LIBERO/SimplerEnv and real-robot tasks. Relative to prior tokenizers, FAST [1] attains the lowest reconstruction error but at a lower compression ratio, while BEAST [2] achieves higher compression with worse reconstruction; OmniSAT sits between these, effectively combining spline alignment with residual VQ. Ablations probe residual layer depth, codebook size, the role of the commitment loss/layer dropout, and the contribution of the spline-based Consistency Encoding itself. The paper also includes ablations backbone model and vision loss but these do not seem to be relevant for the key message of the paper.

[1] Fast: Efficient action tokenization for vision-language-action models

[2] BEAST: Efficient Tokenization of B-Splines Encoded Action Sequences for Imitation Learning

**Strengths:**

- **Comprehensive ablations**: tokenizer hyperparameter ablations, along with isolating contributions from both stages.

- **Real world results**: experiments demonstrating the method on real world setups which adds credibility

- **Reproducibility**: many implementation details are provided (tokenizer hyperparameters, training settings), improving clarity.

**Weaknesses:**

- **Limited novelty (integration of prior components)**: The spline-based fixed-length encoding is closely aligned with prior B-spline control-point approach [1]; residual VQ for action tokenization is also established [2]. The paper mainly combines them with minor implementation changes.

- **AR premise is not fully justified for real-robot control**: The paper claims *Diffusion excels at continuous modeling but scales poorly due to costly denoising. By contrast, AR training on discrete tokens enables efficient optimization and flexible sequence construction, scaling well to heterogeneous data*.  AR action heads trained with cross-entropy (CE) are brittle for real world control, exhibiting unpredictable bin prediction switches with little observation noise. CE loss is non-metric, it only cares about the probability on the ground-truth bin, no matter how the remaining probability mass is distributed among nearby vs far bins. Continuous policies avoid this by design as they directly penalize errors proportional to deviation.

- **Flow/Diffusion is not slow**: The paper’s premise that diffusion/flow models are too slow is questionable. OpenVLA-OFT [3] achieves high throughputs with continuous actions via parallel decoding and action chunking for diffusion models, and even higher throughput by simply switching to a L1 action decoder. Moreover, Real-Time Chunking (RTC) [4] provides asynchronous, look-ahead execution for flow/diffusion VLAs without retraining, directly addressing latency.

- **Fairness of real-robot comparison**: The main result figure should exclude vision-prediction losses. The paper jointly trains an AR objective that predicts future visuals and actions, and the ablation indicates the vision loss contributes substantially to performance. Prior work (e.g., GR-2 [5], UVA [6]) has shown that future-state/video prediction alone can strongly boost policy quality. As a result, the reported gains do not isolate the benefit of the action tokenizer and unfairly advantage the method over baselines that do not use future-state prediction.

[1] BEAST: Efficient Tokenization of B-Splines Encoded Action Sequences for Imitation Learning

[2] Behavior generation with latent actions

[3] Fine-tuning vision-language-action models: Optimizing speed and success

[4] Real-Time Execution of Action Chunking Flow Policies

[5] GR-2: A Generative Video-Language-Action Model with Web-Scale Knowledge for Robot Manipulation

[6] Unified Video Action Model

**Questions:**

- **AR vs Continuous decoder**: On the same data, compare AR approach to (a) a continuous L1 action decoder (OpenVLA-OFT) and (b) a flow-matching decoder ($\pi$0/$\pi$0.5).
- **Codebook behavior**: Report codebook utilization (entropy per layer, dead entries) and how layer-drop affects it.
- **Fair real-robot comparison**: Report an action-only training setting (no visual loss) for your method in the main figure.
- **Revised Introduction/Related Works**: Need to re-position this work appropriately in context of recent developments in continuous control approaches and clarify where AR offers advantage (if any)
- **Latency/Control Frequency Analysis**: Report inference latency, supported control frequencies for AR and continuous control approaches to provide empirical evidence that diffusion denoising is slow.

---

### Official Review · Reviewer_N2Wd · 2025-10-26

**Soundness:** 2
**Presentation:** 2
**Contribution:** 2
**Rating:** 2
**Confidence:** 2

**Summary:**

This paper introduces Omni Swift Action Tokenizer (OmniSAT), a method that improves the efficiency of action sequence compression in Vision-Language-Action (VLA) models. By using B-Spline encoding and multi-stage residual quantization, OmniSAT significantly reduces sequence length (by 6.8×) and lowers target entropy while maintaining reconstruction quality. It also incorporates cross-embodiment learning, leveraging both robot and human demonstrations to enhance scalability and model performance across real-robot and simulation tasks.

**Strengths:**

1. I agree that the exploration of the action tokenizer is valuable.

2. This paper demonstrates the effectiveness on some benchmarks.

**Weaknesses:**

1. The claim in the introduction is wried without any evidence. Please refer to the questions. This has raised some fundamental academic concerns.

2. "Under review as a conference paper at ICLR 2025".. The template is wrong!

3. I did not get the core benefits of the the proposed tokenizer with Faster Tokenizer, and others.  Meanwhile, what's the connection between the tokenizer and the dexego datasets in this paper.

4. I did not get why we need high-quality Compression, it is fewer tokens or more representative representations. For Table 3, this paper ignore the state-of-the-art approaches, e.g., pi0.

**Questions:**

1. ``Diffusion excels at continuous modeling but scales poorly due to costly denoising. By contrast, although AR training operates on discrete tokens and thus cannot directly parameterize the continuous action space, it enables efficient optimization and flexible sequence construction. These properties scale well with heterogeneous data sources, yielding a reliable and robust VLA backbone.'' Diffusion-based approaches scale poorly. Do you have any evidence?

---

### Official Review · Reviewer_TZTB · 2025-11-01

**Soundness:** 3
**Presentation:** 2
**Contribution:** 2
**Rating:** 6
**Confidence:** 3

**Summary:**

This paper proposes an Omni Swift Action Tokenizer, compressing the raw action sequences to a learned representation for efficient training of transformer-based VLAs. Several techniques, such as quantization and VAE, are used for compression without losing important information. Experiments on both simulation and real environments demonstrate the efficacy of the proposed method.

**Strengths:**

- This paper proposes a new tokenizer for better compression of action sequences without losing success rates, which is important for the robotics community, especially considering the large-scale VLAs.
- Empirical results are provided across diverse tasks, and includes both simulation and real environments.

**Weaknesses:**

- For Table 1, the proposed tokenizer exhibits higher MAE than FAST, while no analysis is provided regarding this point.
- While this paper demonstrates that FAST has weak generalization performance, the comparison on o.o.d. generalization seems to be missing.

**Questions:**

- What's the computation required for this method? Is it sample efficiency? Is it easy to reproduce for the community?
- Does the proposed method need fine-tuning for adapting to specific downstream tasks? Where are the empirical results to demonstrate the advantages of the proposed method for cross-embodiment generalization?

---

### Official Review · Reviewer_xJse · 2025-11-05

**Soundness:** 3
**Presentation:** 3
**Contribution:** 2
**Rating:** 4
**Confidence:** 3

**Summary:**

The paper proposes OmniSAT, an action tokenizer for vision-language-action (VLA) policies that aims to compress continuous robot action trajectories into discrete tokens to accelerate autoregressive (AR) training while preserving execution fidelity. The method has two stages: (1) “consistency encoding” that normalizes per-DoF ranges and aligns variable-length trajectories using B-spline control points; (2) multi-stage residual vector quantization (RVQ) applied separately to position, rotation, and gripper groups to produce compact discrete indices.
The tokenizer is pretrained on DROID and then used for AR policy training; a cross-embodiment strategy is used to mix robot and human demonstrations in the unified token space.
Experimental results show that it can acheive ~6.8x end-to-end compression with low reconstruction error, faster convergence, and state-of-the-art (or competitive) success rates on LIBERO, SimplerEnv, and three real-robot tasks.

**Strengths:**

- This paper tackles a important bottleneck in AR-based VLA training and propose a two-stage tokenizer which looks reasonable to me.
- The paper is well-written and easy to follow
- The performance looks good, it achieves 47%(6.8/4.6) and 83%(6.8/3.7) relative improvement on compression ratio compared to BEAST and FAST baselines and achieves better task success rate on sim and real benchmarks.

**Weaknesses:**

- The authors claim "shorter sequences and lower target entropy" but I do not see corresponding metrics in the paper.
- Overall I feel like the tokenizer improvement acts as a black box as several metrics are not reported (see Questions sec.) in this paper. Which makes it hard to understand why such improvement is principled compared to the baseline and to what circumstances the improvement in tokenizer can be transferred to end-to-end task performance boost.

**Questions:**

- BEAST has the worst MAE but does not fall behind OmniSAT and even surpass FAST on performance wise. I wonder what's the authors' comment on this situation and whether reconstruction quality / compression ratio truly reflect the tokenizer's quality.
- Baseline details are missing, did the authors also fine-tune the baseline tokenizers?
- Would be great if the authors could provide a in-depth error analysis about the tokenizer performance (sequence length, entropy, reconstruction metrics for each DoF, etc.) and study the correlations between the tokenizer performance and the final model performance to help the readers better understand and make improvement on this direction.
- I also wonder how generalizable the trained tokenizer is when training it on one dataset and deploy it on another, and whether the convergence speed is also faster on other benchmark (SimplerEnv / real world exp) for the proposed tokenizer?
- The authors claim a short sequence length due to higher compression ratio, which should also transfer to the running time since AR model can predict less token to complete the task and it should be one of the main benefit. Would be great if the authors could also report this metric.

---

### Note · Authors · 2025-11-20

I have read and agree with the venue's withdrawal policy on behalf of myself and my co-authors.